# Analysis of Chicken IFITM3 Gene Expression and Its Effect on Avian Reovirus Replication

**DOI:** 10.3390/v16030330

**Published:** 2024-02-21

**Authors:** Hongyu Ren, Sheng Wang, Zhixun Xie, Lijun Wan, Liji Xie, Sisi Luo, Meng Li, Zhiqin Xie, Qing Fan, Tingting Zeng, Yanfang Zhang, Minxiu Zhang, Jiaoling Huang, You Wei

**Affiliations:** 1Guangxi Key Laboratory of Veterinary Biotechnology, Guangxi Veterinary Research Institute, Nanning 530000, China; renhongyu328@126.com (H.R.); wangsheng1021@126.com (S.W.); wanlijun0529@163.com (L.W.); xie3120371@163.com (L.X.); 2004-luosisi@163.com (S.L.); mengli4836@163.com (M.L.); xzqman2002@sina.com (Z.X.); fanqing1224@126.com (Q.F.); tingtingzeng1986@163.com (T.Z.); zhangyanfang409@126.com (Y.Z.); zhminxiu2010@163.com (M.Z.); huangjiaoling728@126.com (J.H.); weiyou0909@163.com (Y.W.); 2Key Laboratory of China (Guangxi)-ASEAN Cross-Border Animal Disease Prevention and Control, Ministry of Agriculture and Rural Affairs of China, Nanning 530000, China

**Keywords:** IFITM3, avian reovirus, bioinformatics analysis, antiviral, innate immunity

## Abstract

Interferon-inducible transmembrane protein 3 (IFITM3) is an antiviral factor that plays an important role in the host innate immune response against viruses. Previous studies have shown that IFITM3 is upregulated in various tissues and organs after avian reovirus (ARV) infection, which suggests that IFITM3 may be involved in the antiviral response after ARV infection. In this study, the chicken IFITM3 gene was cloned and analyzed bioinformatically. Then, the role of chicken IFITM3 in ARV infection was further explored. The results showed that the molecular weight of the chicken IFITM3 protein was approximately 13 kDa. This protein was found to be localized mainly in the cytoplasm, and its protein structure contained the CD225 domain. The homology analysis and phylogenetic tree analysis showed that the IFITM3 genes of different species exhibited great variation during genetic evolution, and chicken IFITM3 shared the highest homology with that of *Anas platyrhynchos* and displayed relatively low homology with those of birds such as *Anser cygnoides* and *Serinus canaria*. An analysis of the distribution of chicken IFITM3 in tissues and organs revealed that the IFITM3 gene was expressed at its highest level in the intestine and in large quantities in immune organs, such as the bursa of Fabricius, thymus and spleen. Further studies showed that the overexpression of IFITM3 in chicken embryo fibroblasts (DF-1) could inhibit the replication of ARV, whereas the inhibition of IFITM3 expression in DF-1 cells promoted ARV replication. In addition, chicken IFITM3 may exert negative feedback regulatory effects on the expression of TBK1, IFN-γ and IRF1 during ARV infection, and it is speculated that IFITM3 may participate in the innate immune response after ARV infection by negatively regulating the expression of TBK1, IFN-γ and IRF1. The results of this study further enrich the understanding of the role and function of chicken IFITM3 in ARV infection and provide a theoretical basis for an in-depth understanding of the antiviral mechanism of host resistance to ARV infection.

## 1. Introduction

Avian reovirus (ARV) is a pathogen that circulates widely in poultry and can cause viral arthritis, tenosynovitis and malabsorption syndrome. This virus can also induce severe immunosuppression, which can easily lead to complications or secondary infection with other diseases. These effects lead to reduced production performance and increased mortality in chickens [1,2,3], resulting in major economic losses in the poultry industry. At present, the prevention and control of ARV mainly involve vaccination, but due to continuous mutation of the strain, the expected immune protection effect is not achieved [4,5,6]. Therefore, in-depth study of the innate immune regulatory mechanism of ARV infection is highly important for its prevention and control.

Innate immunity is the body’s first line of defense against viral infections. After viral invasion, the pattern recognition receptor of the host cell specifically recognizes the molecular pattern associated with the pathogen and thereby activates specific signaling pathways and induces the production of antiviral cytokines such as interferon (IFN) and interleukin, which causes the body to enter an antiviral state [7,8]. Among these, the interferon-mediated antiviral effect is an important part of the host antiviral response [9]. IFN induces the production of many interferon-stimulated genes (ISGs) by activating the JAK/STAT signaling pathway [10], and ISGs are the main executors of IFN antiviral functions [11]. Studies have shown that ARV infection can induce the transcriptional expression of IFN-α, IFN-β and ISGs such as IFITM1, IFITM3, IFIT5, Mx, ISG12 and other cytokines in various tissues and organs; at the early stage of ARV infection, IFITM3 is significantly upregulated in peripheral blood lymphocytes, joints, the thymus and the bursa of Fabricius and shows a consistent trend with the viral load of ARV in these tissues and organs, which suggests that IFITM3 plays a crucial role in ARV infection [12,13,14].

The IFITM is an important ISG. Different species have different varieties of IFITM proteins. The human IFITMs include IFITM1-3, IFITM5 and IFITM10. Similar to humans, chickens also have five IFITM genes. IFITMs play a significant role in biological activities such as tumorigenesis, cell adhesion and immune signal transduction [15,16]. According to previous research findings, IFITM1, IFITM2 and IFITM3 are related to immune regulatory processes in the body, and the expression of IFITM3 exerts a certain limiting effect on the replication of a variety of highly pathogenic viruses [17,18,19]. IFITM3 is expressed in fish, amphibians, poultry and mammals, and its antiviral activity is relatively conserved from prokaryotes to vertebrates [20,21]. Early studies have found that IFITM3 can effectively inhibit the replication of a variety of enveloped viruses, such as influenza A viruses (IAVs), dengue virus (DENV) and Ebola virus (EBOV) [21,22,23], mainly by affecting the fusion of the virus to the endosomal membrane to prevent the virus from entering cells [24,25]. Additional studies in this research area revealed that IFITM3 effectively inhibits nonenveloped viruses, such as foot-and-mouth disease virus (FMDV), norovirus (NoV) and mammalian orthoreoviruses [18,26,27]. Although nonenveloped viruses cannot mediate membrane fusion via proteins on the envelope as do enveloped viruses, they still need to enter cells through endosomes. IFITM3 inhibits viral replication by inhibiting the process of virus entry from endosomes into the cytoplasm. The antiviral mechanism of IFITM3 in mammalian orthoreovirus infection has been well described, and IFITM3 restricts the entry of the virus into host cells by altering the acidic environment of endosomes and reducing protease activity [18]. However, the mechanism of action of chicken IFITM3 has been relatively poorly studied, and the role of IFITM3 in ARV infection has not been reported.

Therefore, in this study, we conducted further investigations on the biological role of chicken IFITM3 in preventing ARV infection. First, we cloned the chicken IFITM3 gene and performed bioinformatic analysis and analyses of its subcellular localization and tissue-organ distribution. Subsequently, the effect of IFITM3 on ARV replication and the regulatory effect of IFITM3 on the expression of correlated molecules in the innate immune signaling pathway were analyzed via overexpression or RNA inhibition assays. The results of this study will provide new ideas for further exploration of the mechanism of the innate immune response to host resistance during ARV infection.

## 2. Materials and Methods

### 2.1. Ethics Statement

This study was approved by the Animal Ethics Committee of Guangxi Veterinary Research Institute. The animal experiments and sample collection were conducted in accordance with the guidance of protocol #2019C0408 issued by the Animal Ethics Committee of Guangxi Veterinary Research Institute.

### 2.2. Animals, Virus and Cells

The “white leghorn” specific-pathogen-free (SPF) chicken embryos used in this study were purchased from Beijing Boehringer Ingelheim Vital Biotechnology Co., Ltd. (Beijing, China). The ARV S1133 strain was purchased from the China Institute of Veterinary Drug Control. DF-1 cells were preserved in our laboratory and cultured in DMEM (Gibco, Grand Island, NY, USA) supplemented with 10% fetal bovine serum (Gibco).

### 2.3. Cloning and Bioinformatics Analysis of the IFITM3 Gene

The nucleotide sequences of IFITM3 genes were downloaded from the National Center for Biotechnology Information (NCBI) database. Sequence alignment analysis was performed with DNAstar 7.1 to design primers (Table 1). Total RNA was extracted from DF-1 cells and reverse-transcribed into cDNA, and the resulting cDNA was subsequently used as a template for the amplification of the IFITM3 gene.

The conserved domain was predicted based on the NCBI CD-Search database. SOPMA software (http://npsa-pbil.ibcp.fr/cgi-bin/npsa_automat.pl?page=npsa_sopma.html, accessed on 17 February 2024) was used for secondary structure prediction analysis of the chicken IFITM3 protein. SWISS-MODEL (https://swissmodel.expasy.org/interactive#alignment, accessed on 20 February 2024) was used to predict the tertiary structure of the protein. DNAstar 7.1 and MEGA 11 were used for homology analysis and phylogenetic tree construction. The GenBank accession numbers of the IFITM3 genes from different species are shown in Table 2.

### 2.4. Overexpression of the IFITM3 Protein

The recombinant plasmid pEF1α-Myc-IFITM3 was constructed and transfected into DF-1 cells using Lipofectamine^TM^ 3000 (Invitrogen, Carlsbad, CA, USA) to overexpress the IFITM3 protein. Twenty-four hours after transfection, the cells were infected with ARV S1133 (MOI = 1), and cell samples and culture medium supernatant were collected 24 h later. RNA from cell samples was extracted and reverse-transcribed into cDNA. The changes in the expression of ARV σC gene and innate immune signaling pathway-correlated molecules were detected by real-time fluorescence quantitative PCR (RT–qPCR). The utilized primers [12,14,28] were described previously (Table 1). In addition, the abovementioned culture medium was diluted for the infection of DF-1 cells. The lesions of the cells were observed and recorded, and the TCID_50_ of the virus was calculated by the Reed–Muench method.

### 2.5. IFITM3 RNA Interference Assay

Three small interfering RNAs (siRNAs) for the chicken IFITM3 gene were designed (Table 3), and the utilized primers were synthesized by GenePharma (Suzhou, China). siRNAs or siNCs (30 pmol) were transfected separately into DF-1 cells using LipofectamineTM RNAiMAX (Invitrogen) to inhibit the expression of IFITM3 protein. Twenty-four hours after transfection, the cells were infected with the ARV S1133 strain, and cell samples and culture medium supernatant were collected 24 h later. The cell samples were used to detect the changes in the expression of ARV σC gene and innate immune signaling pathway-correlated molecules, and the culture supernatant was used for the detection of viral replication.

### 2.6. RNA Extraction and RT–qPCR

Total RNA was extracted from the samples using a TRIzol kit (Invitrogen). The RNA was reverse-transcribed to cDNA using Maxima™ H Minus cDNA Synthesis Master Mix (Thermo Fisher Scientific, Boston, MA, USA) and stored at −80 °C for subsequent assays.

Based on the gene sequence information in GenBank, primers for ARV σC, IFITM3 and innate immune signaling pathway-related molecules were designed and synthesized (Table 1). RT–qPCR was performed using PowerUp SYBR Green Master Mix (Thermo Fisher Scientific), and the GAPDH gene served as an internal control. The reaction program was as follows: 94 °C for 2 min and 40 cycles of 94 °C for 15 s and 60 °C for 30 s. The detection results were analyzed by the 2^−ΔΔCt^ method.

### 2.7. Confocal Microscopy Analysis of the Subcellular Localization of the IFITM3 Protein

The cells were transfected with the recombinant plasmids pEF1α-Myc-IFITM3 and pEF1α-Myc, respectively. After 24 h of incubation, the culture medium was discarded. The cells were subsequently washed three times with phosphate-buffered saline (PBS) (Solarbio, Beijing, China) and fixed with 4% paraformaldehyde (Solarbio) for 30 min at room temperature. After three washes with PBS, the cells were infiltrated with 0.1% Triton X-100 (Solarbio) for 15 min and blocked with 5% BSA (Solarbio) for 1 h at room temperature. The cells were incubated with mouse anti-Myc monoclonal antibody (Invitrogen) as the primary antibody at 37 °C for 2 h and then with Alexa Fluor 488-labeled goat anti-mouse IgG (Invitrogen) as the secondary antibody at 37 °C while protected from light for 1 h. The nuclei were then stained with DAPI (Solarbio) for 10 min at room temperature while protected from light. After washing with PBS, 50% glycerol was added to the cell plates, and the results were observed by laser confocal microscopy.

### 2.8. Western Blotting

The cells transfected with the recombinant plasmids were washed with PBS and lysed on ice for 30 min using lysis buffer supplemented with protease inhibitors (Sangon Biotech, Shanghai, China). The lysate was then boiled at 100 °C for 10 min and centrifuged to obtain protein samples. The proteins were separated by SDS–PAGE and then transferred to polyvinylidene difluoride membranes (Millipore, Billerica, MA, USA). The membranes were blocked overnight with 5% skim milk at 4 °C and incubated with primary antibody at 37 °C for 2 h and then with the secondary antibody for 1 h. Mouse anti-Myc monoclonal antibody (Invitrogen) and mouse anti-β-actin antibody (Invitrogen) were used as primary antibodies. AP-labeled goat anti-mouse IgG (H+L) (Beyotime Biotechnology, Beijing, China) was used as the secondary antibody. The proteins were then visualized using a BCIP/NBT alkaline phosphatase color development kit (Beyotime Biotechnology).

### 2.9. Statistical Analysis

All the data were statistically analyzed using Student’s t test and graphed using GraphPad Prism 8. The data were obtained from biological replicates and technical replicates. The results are expressed as the mean ± standard deviation (SD) of three independent experiments. Each sample was measured three times during RT–qPCR. * indicates *p* < 0.05, ** indicates *p* < 0.01, *** indicates *p* < 0.001, and **** indicates *p* < 0.0001.

## 3. Results

### 3.1. Cloning, Bioinformatics Analysis and Subcellular Localization of IFITM3

The full-length sequence of chicken IFITM3 (approximately 342 bp) was successfully cloned using the IFITM3-1 primers (Figure 1). The sequence was uploaded to the NCBI-BLAST online website for comparison, and the results confirmed that the cloned sequence was the full-length sequence encoded by the IFITM3 gene of *Gallus gallus*, which consists of 342 bases and encodes a total of 113 amino acids. Based on the NCBI CD-Search, the CD225 conserved domain in the chicken IFITM3 protein was predicted (Figure 2A). The secondary structure analysis of the IFITM3 protein showed that alpha helices accounted for 42.48%, beta turns accounted for 1.77%, random coils accounted for 40.71%, and extended strands accounted for 15.04% (Figure 2B). Tertiary structure prediction showed that the global model quality estimation (GMQE) of the chicken IFITM3 protein and IFITM3 derived from Northern Bobwhite equaled 0.61, and the coverage rate was 80.91% (Figure 2C). Homology analysis revealed that chicken IFITM3 exhibited the highest homology (99.4%) with that of *Anas platyrhynchos*. The homologies between chicken IFITM3 and those of *Anser cygnoides* and *Serinus canaria* were 46% and 45.7%, respectively, and the homologies between chicken IFITM3 and those of *Homo sapiens*, *Gorilla gorilla gorilla*, *Capra hircus*, *Sus scrofa* and *Mus musculus* were 50.4%, 53.5%, 51%, 49% and 46.5%, respectively. We constructed a phylogenetic tree to explore the genetic relationships between chicken IFITM3 and IFITM3s from other species (Figure 3). The results showed that chicken IFITM3 is most closely related to IFITM3 in *A. platyrhynchos*. *A. cygnoides* and *S. canaria* are found in the same group of birds as chickens, but their IFITM3s are distantly related to chicken IFITM3. Chicken IFITM3 is most distantly related to IFITM3s in mammals, such as *H. sapiens* and *G. gorilla gorilla*. These results are consistent with the results of the homology analysis described above. The subcellular localization of the IFITM3 protein in DF-1 cells was analyzed by immunofluorescence and laser confocal microscopy. The nuclei were labeled with blue fluorescence, and the IFITM3 protein was labeled with green fluorescence. As shown in Figure 4, DF-1 cells transfected with the pEF1α-Myc-IFITM3 plasmid exhibit green fluorescence in the cytoplasm, whereas control cells transfected with the pEF1α-Myc vector do not show green fluorescence, indicating that the IFITM3 protein is localized in the cytoplasm of DF-1 cells.

### 3.2. Distribution Characteristics of IFITM3 in Chicken Tissues and Organs

The distribution of the IFITM3 gene in different tissues and organs of 14-day-old SPF chickens was determined by RT–qPCR. The results showed that IFITM3 was widely expressed in a variety of tissues and organs of chickens, and its highest expression was found in the intestine, followed by the bursa of Fabricius, blood, lung, pancreas, trachea, thymus and spleen. IFITM3 was expressed at low levels in the liver, skin, heart, glandular stomach, gizzard, joint and kidney, and its relative expression in muscle and brain tissues was extremely low (Figure 5).

### 3.3. High-Level Expression of IFITM3 Reduces ARV Replication

The effect of IFITM3 on ARV replication was analyzed via overexpression and interference assays. First, the overexpression of chicken IFITM3 in DF-1 cells was verified by Western blotting and RT–qPCR. Western blot analysis of DF-1 cell samples transfected with pEF1α-Myc-IFITM3 revealed that a specific band of approximately 13 kDa could be detected by Myc-tagged antibody, whereas no specific bands were detected in cell samples transfected with the empty vector (Figure 6A). The RT–qPCR results showed that, compared with that in the control group, the expression of IFITM3 in D-F1 cells transfected with pEF1α-Myc-IFITM3 was significantly upregulated, and its expression increased by approximately 155-fold (Figure 6B). Subsequently, IFITM3 was overexpressed in DF-1 cells, the resulting cells were subsequently infected with ARV, and the mRNA level of the ARV σC gene was then detected by RT–qPCR to determine the changes in the viral load. The results showed that the viral load of ARV was significantly reduced after IFITM3 overexpression (Figure 6C). Moreover, the detection of viral titers in cell culture supernatants also showed that the viral titer of ARV after the overexpression of IFITM3 was significantly lower than that in the control group (Figure 6D). Therefore, we inferred that the overexpression of chicken IFITM3 could effectively inhibit the replication of ARV, and based on this finding, we speculated that inhibition of the expression of IFITM3 may be beneficial for ARV replication. In the following experiments, three siRNAs were designed and synthesized to inhibit the expression of IFITM3. As shown in Figure 6E, si242 exerted the greatest inhibitory effect. The expression of IFITM3 in DF-1 cells was inhibited by transfection with si242, and ARV infection was performed 24 h after transfection. The mRNA level of the ARV σC gene and the viral titer in the cell supernatant were subsequently measured. The results showed that the level of ARV replication increased significantly after inhibition of the expression of IFITM3 (Figure 6F,G). The results were consistent with the expectations.

### 3.4. Effect of IFITM3 on Innate Immune Signaling Pathway-Correlated Molecules during ARV Infection

The above-described test results showed that chicken IFITM3 exerts an inhibitory effect on the replication of ARV. To further explore the antiviral mechanism of IFITM3 in the process of ARV infection, we studied the regulatory effect of IFITM3 on the innate immune response after ARV infection. IFITM3 was overexpressed or inhibited in DF-1 cells, and 24 h later, the cells were infected with ARV. The changes in the expression of molecules related to the innate immune signaling pathway were then detected by RT–qPCR, and the results are shown in Figure 7. After infection, the expression of MAVS, IRF7, STING, NF-κB, MAD5, LGP2, IFN-α and IFN-β was upregulated compared with that in the control group, regardless of whether IFITM3 was overexpressed or inhibited. Interestingly, the expression levels of IRF1, TBK1 and IFN-γ were significantly downregulated after IFITM3 overexpression (*p* < 0.05 or *p* < 0.01). However, the expression levels of IRF1, TBK1 and IFN-γ were significantly upregulated after IFITM3 inhibition (*p* < 0.05 or *p* < 0.001). It is speculated that changes in the expression of IFITM3 during ARV infection may affect the expression of IRF1, TBK1 and IFN-γ.

## 4. Discussion

IFITM3 is an important effector of the innate immune system and plays an important role in host resistance to viral infections. Gene structure is often closely related to biological function. In this study, we cloned the chicken IFITM3 gene and analyzed this gene via bioinformatic approaches. Multiple comparison analyses revealed that the homology between chicken IFITM3 and *A. platyrhynchos* was as high as 99.4%. Furthermore, chicken IFITM3 did not exhibit more than 50% homology with that of *A. cygnoides* or *S. canaria*. The homologies of chicken IFITM3 with those of other mammals did not exceed 55%. The same results were obtained via phylogenetic tree analysis, which revealed substantial genetic variation in the IFITM3 genes of different species. The homology between chicken IFITM3 and *A. platyrhynchos* IFITM3 was high, whereas chicken IFITM3 exhibited low homology with bird IFITM3s, such as those of *A. cygnoides* and *S. canaria*. Studies have shown that IFITM proteins belong to the CD225 superfamily, and their members share a highly conserved region of amino acids, the CD225 domain [29]. The CD225 domain was also identified in the protein structure analysis of chicken IFITM3. The CD225 domain consists of an intramembrane domain (IMD), cytoplasmic intracellular loop (CIL) and transmembrane domain (TMD) [30]. Previous studies have shown that the CD225 domain contains multiple key regions associated with antiviral effects, which are also closely correlated with the antiviral effects of IFITM3 [31]. The first intramembrane domain (IM1) contains two critical residues, F75 and F78, which are decisive factors affecting the interaction of the IFITM3 protein with the host [29]. GxxxG is an oligomeric motif in the CD225 domain. The glycine-95 in GxxxG is closely related to the oligomerization of IFITM3 and its antiviral activity [32]. Therefore, the conserved structure of the CD225 superfamily may cause IFITM3 proteins derived from different species to exhibit certain similarities in their biological functions. Additionally, the question of whether IFITM3 proteins from different species have antiviral specificity deserves in-depth study, as do their mechanisms of action.

The antiviral function of proteins is closely related to their subcellular localization in cells and their distribution in tissues and organs. The subcellular localization of the chicken IFITM3 protein in DF-1 cells was analyzed by laser confocal microscopy, and the results showed that IFITM3 was localized in the cytoplasm of DF-1 cells. Some previous analyses have investigated the subcellular localization of the IFITM3 protein. S. E. Smith [33] reported that the IFITM3 protein in chickens localizes to the perinuclear area of DF-1 cells, and the human IFITM3 protein also localizes to the perinuclear area of human-derived A549 cells. The distribution of IFITM3 in chicken tissues and organs was then analyzed, and the highest expression of the IFITM3 gene was found in the chicken intestine. This expression pattern is similar to that of human IFITM3, which is most highly expressed in the ileum and cecum in the human digestive system [34]. Moreover, IFITM3 is significantly upregulated in the intestines of pigs infected with porcine circovirus type 2 (PCV2) and porcine parvovirus virus (PPV) [35]. Zoya Alteber et al. [15] experimented with IFITM3-deficient mice and revealed that the IFITM3 gene is involved in regulating the stability of the intestinal environment. IFITM3 also significantly improves the incidence of colitis and prevents inflammation-associated tumorigenesis. After ARV infection, the virus replicates primarily in the host’s gut and subsequently spreads through the fecal-oral route and respiratory tract [1,36,37]. However, whether ARV infection further induces the expression of the IFITM3 gene in the gut is unknown. Furthermore, IFITM3 was found to be abundantly expressed in immune organs such as the bursa of Fabricius, thymus and spleen. Studies have shown that high expression of IFITM3 can be induced in a variety of immune organs after ARV infection, which is generally consistent with the trend found for the expression of ARV [12,14].

Further experimental results showed that overexpression of IFITM3 in DF-1 cells could inhibit the replication of ARV, whereas the inhibition of IFITM3 expression in DF-1 cells could promote the replication of ARV, indicating that IFITM3 is an important antiviral factor against ARV infection. Recent studies have shown that IFITM3 can also inhibit the replication of a variety of avian-derived viruses. Stable expression of the duck IFITM3 protein in DF-1 cells can significantly limit the replication of H6N2 and H11N9 IAV strains [38]. Both chicken and duck IFITM3 can effectively inhibit the replication of avian Tembusu virus (ATMUV) [39].

To date, studies on the antiviral mechanism of IFITM proteins have focused mainly on their ability to block contact between viruses and cells and less on their ability to regulate innate immune signaling pathways. Therefore, in this study, the effect of IFITM3 on the expression of innate immune-related molecules after ARV infection was further investigated. The results showed that the expression of MAVS, IRF7, STING, NF-κB, MAD5, LGP2, IFN-α and IFN-β was upregulated compared with that in the control group, regardless of whether IFITM3 was overexpressed or inhibited after infection. Previous studies have shown that ARV infection induces the upregulation of these cytokines [12]. It is hypothesized that changes in IFITM3 expression during ARV infection may not affect the expression of these molecules. In addition, previous studies revealed that the expression of TBK1 and IFN-γ is significantly upregulated after ARV infection [12]. However, in the present study, the expression of TBK1, IFN-γ and IRF1 was significantly downregulated after the overexpression of IFITM3 and significantly upregulated after the inhibition of IFITM3. This interesting phenomenon deserves more in-depth discussion. Researchers have shown that IFITM3 is induced by type I IFNs and can also negatively regulate the production of type I IFNs [40], indicating that IFITM3 may play a role in innate immunity as a negative feedback regulator. TBK1 is an important linker molecule that connects upstream receptor signaling and downstream gene activation in apoptosis, inflammation and immune responses [41,42]. TBK1 has been found to be involved in lipopolysaccharide (LPS)-induced IFITM3 expression [43]. This finding suggests a potential link between IFITM3 and TBK1 in the body’s inflammatory response. Nevertheless, in ARV infection, IFITM3 may exert a negative regulatory effect on TBK1. IFN-γ is a cytokine with antiviral activity and immunomodulatory functions that can act on different types of immune cells to regulate innate and adaptive immunity [44]. IRF1 plays a very important role in the innate immune response induced by IFN-γ [45,46]. IFN-γ mainly regulates transcription factors such as IRF1 through the JAK/STAT signaling pathway and thus drives subsequent transcriptional regulation [47]. In this study, the overexpression of IFITM3 during ARV infection significantly downregulated the expression of IFN-γ and IRF1, whereas the inhibition of IFITM3 significantly upregulated the expression of IFN-γ and IRF1. It is hypothesized that IFITM3 may be involved in the body’s innate immune response by negatively regulating IFN-γ and IRF1 during ARV infection.

In this study, the chicken IFITM3 gene was cloned and bioinformatically analyzed, and its role in ARV infection was then further analyzed. The results of this study lay a theoretical foundation for obtaining an in-depth understanding of the antiviral mechanism of host resistance to ARV and provide new ideas for the development of new ARV prevention measures. However, the specific regulatory mechanism of IFITM3 on innate immunity during ARV infection needs to be further studied.

## Figures and Tables

**Figure 1 viruses-16-00330-f001:**
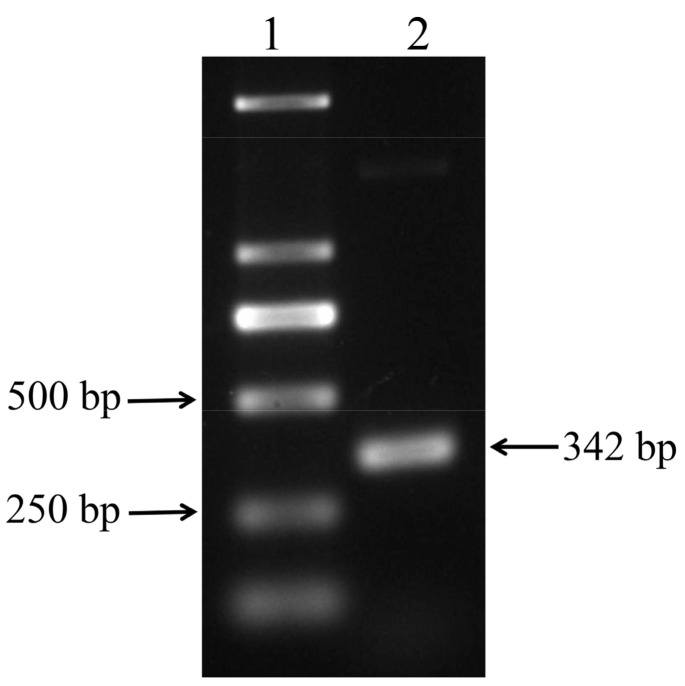
Analysis of the PCR product of the IFITM3 gene via agarose gel electrophoresis. Lane 1: DL2000 DNA marker; Lane 2: Amplification product of the IFITM3 gene. The size of the amplified IFITM3 gene fragment is 342 bp.

**Figure 2 viruses-16-00330-f002:**
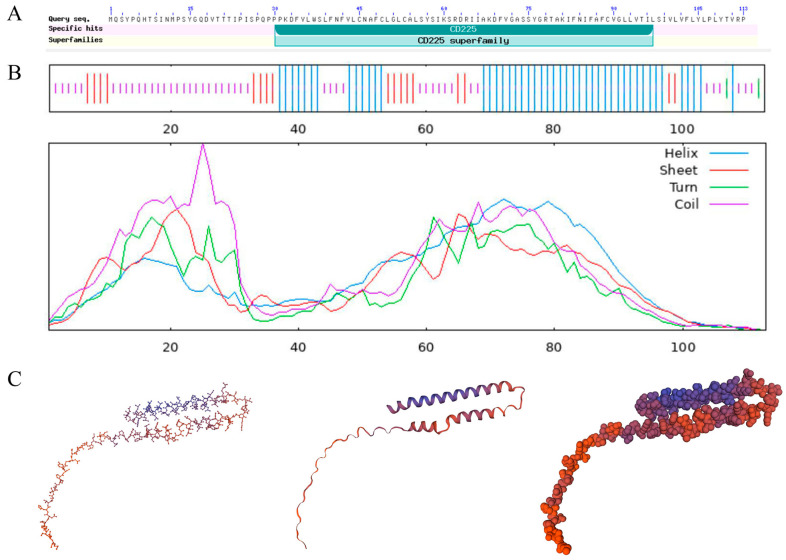
Structural analysis of the IFITM3 protein. (**A**). Schematic diagram of the conserved domain of the IFITM3 protein. The diagram shows the CD225 conserved domain. (**B**). Secondary structure of the IFITM3 protein. The longest lines represent alpha helices (Hh), the second longest lines represent extended strands (Ee), the third longest lines represent beta turns (Tt), and the shortest lines represent random coils (Cc). (**C**). Tertiary structure of the IFITM3 protein. The global model quality estimation (GMQE) of the chicken IFITM3 protein and IFITM3 derived from Northern Bobwhite equaled 0.61, and the coverage rate was 80.91%.

**Figure 3 viruses-16-00330-f003:**
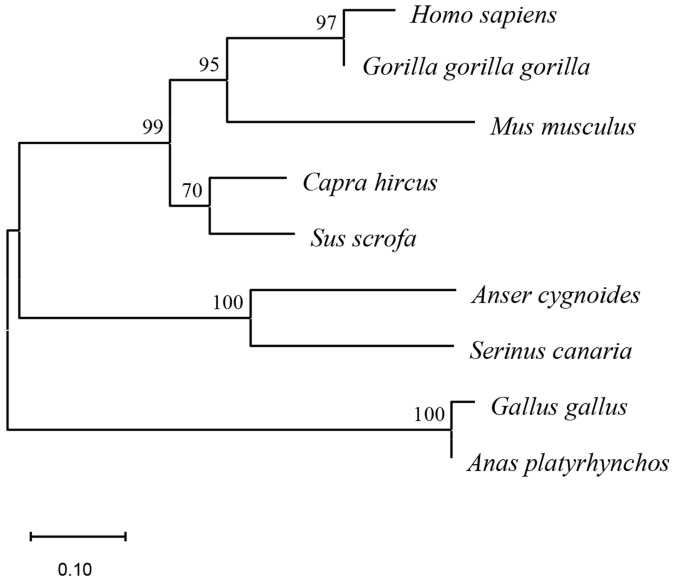
Phylogenetic tree analysis of IFITM3 genes in different species. A phylogenetic tree was constructed using MEGA 11 via the neighbor-joining method. The scale bar indicates the length of the branches, and the bootstrap confidence values are shown on the nodes of the tree.

**Figure 4 viruses-16-00330-f004:**
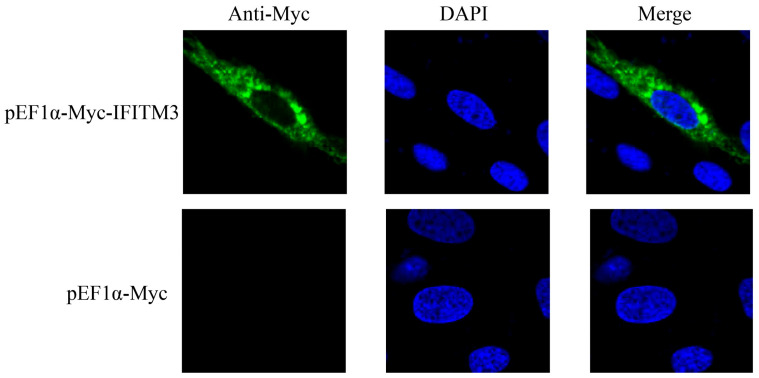
Subcellular localization of the IFITM3 protein (63× magnification). The subcellular localization of the IFITM3 protein in DF-1 cells was observed by laser confocal microscopy. The panels show nuclei stained with DAPI (blue), the IFITM3 protein labeled with Alexa Fluor 488-labeled goat anti-mouse IgG (green) and a merged image.

**Figure 5 viruses-16-00330-f005:**
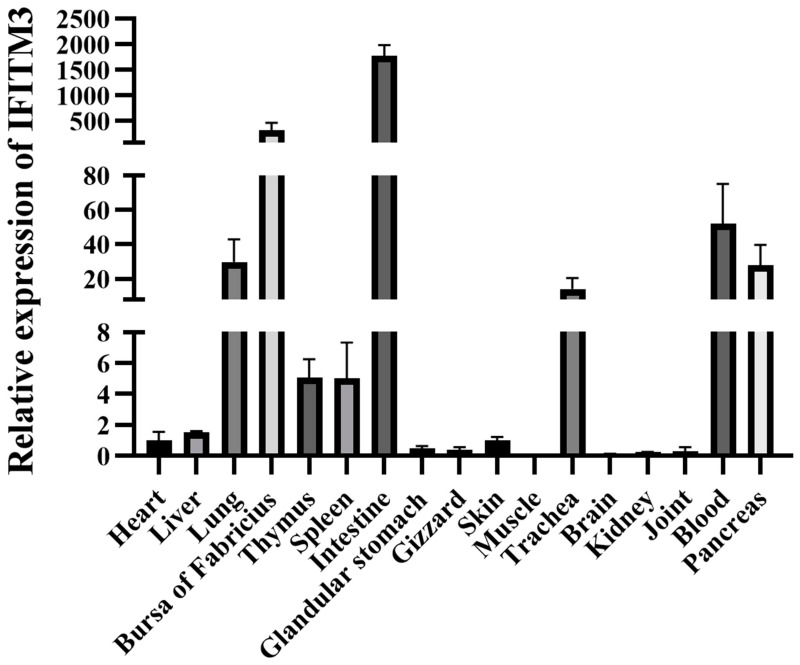
Analysis of the IFITM3 gene expression in different tissues and organs. RT–qPCR was used to measure the IFITM3 mRNA levels in the heart, liver, lung, bursa of Fabricius, thymus, spleen, intestine, glandular stomach, gizzard, skin, muscle, trachea, brain, kidney, joint, blood and pancreas of 14-day-old SPF chickens. The data are presented as the mean ± SD of three independent experiments.

**Figure 6 viruses-16-00330-f006:**
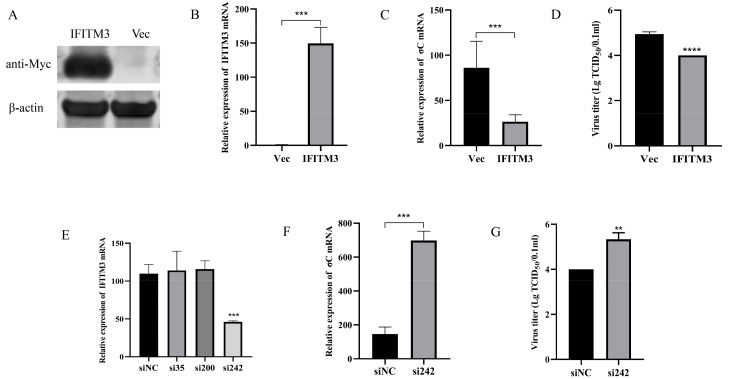
IFITM3 inhibits the replication of ARV in DF-1 cells. DF-1 cells were transfected with pEF1α-Myc-IFITM3 or pEF1α-Myc (Vec), and high levels of IFITM3 expression in DF-1 cells were confirmed by Western blotting (**A**) and RT–qPCR (**B**). DF-1 cells were transfected with pEF1α-Myc-IFITM3 or pEF1α-Myc (Vec) and then infected with the ARV S1133 strain (MOI = 1). After 24 h, the replication of ARV in DF-1 cells was detected by RT–qPCR (**C**) and by determining the viral titer (**D**). The inhibition efficiency of the three siRNAs on IFITM3 was detected by RT–qPCR (**E**). DF-1 cells were transfected with si242 or siNC and then infected with the ARV S1133 strain (MOI = 1). After 24 h, the replication of ARV in DF-1 cells was detected by RT–qPCR (**F**) and by determining the viral titer (**G**). Asterisks indicate significant differences (** *p* < 0.01, *** *p* < 0.001, **** *p* < 0.0001).

**Figure 7 viruses-16-00330-f007:**
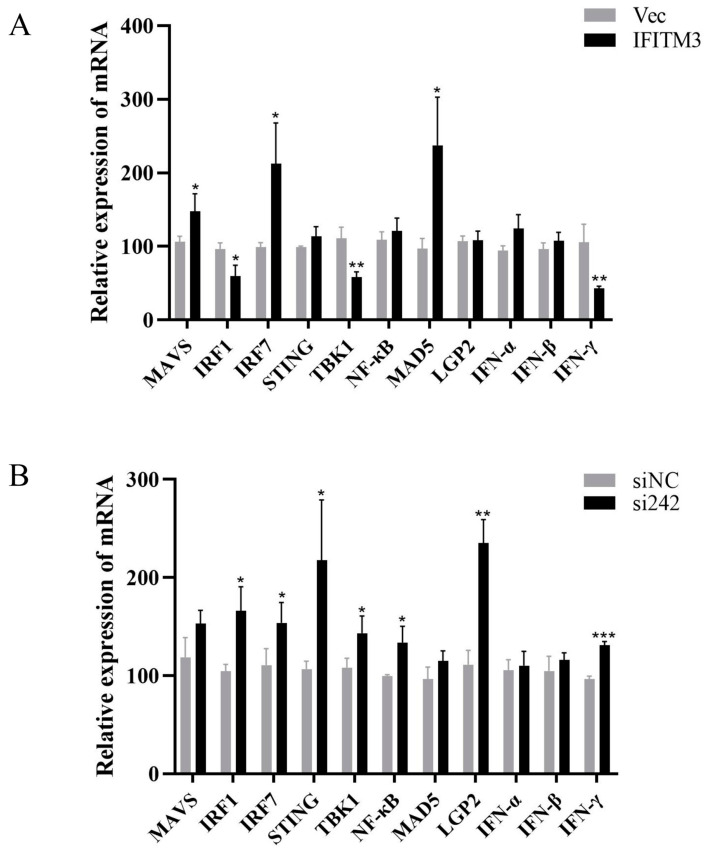
Effect of IFITM3 on the expression of molecules associated with innate immune signaling pathways during ARV infection. DF-1 cells were transfected with pEF1α-Myc-IFITM3 or pEF1α-Myc (Vec) and then infected with the ARV S1133 strain (MOI = 1). After 24 h of infection, RT–qPCR was used to measure the changes in the expression of MAVS, IRF1, IRF7, STING, TBK1, NF-κB, MAD5, LGP2, IFN-α, IFN-β and IFN-γ (**A**). DF-1 cells were transfected with si242 or siNC and then infected with the ARV S1133 strain (MOI = 1). After 24 h of infection, RT–qPCR was used to measure the changes in the expression of MAVS, IRF1, IRF7, STING, TBK1, NF-κB, MAD5, LGP2, IFN-α, IFN-β and IFN-γ (**B**). The data are presented as the mean ± SD of three independent experiments. Asterisks indicate significant differences (* *p* < 0.05, ** *p* < 0.01, *** *p* < 0.001).

**Table 1 viruses-16-00330-t001:** PCR primers used in this study.

Primers	Primer Sequences (5′-3′)	Usage
IFITM3-1	F: GCGTCGACCATGCAGAGCTACCCTCAGCAC R: GCGCGGCCGCTCAGGGCCTCACAGTGTACAA	RT–PCR
IFITM3-2	F: GGAGTCCCACCGTATGAAC R: GGCGTCTCCACCGTCACCA	RT–qPCR
ARV σC	F: CCACGGGAAATCTCACGGTCACT R: TACGCACGGTCAAGGAACGAATGT	RT–qPCR
MAVS	F: CCTGACTCAAACAAGGGAAG R: AATCAGAGCGATGCCAACAG	RT–qPCR
IRF1	F: GCTACACCGCTCACGA R: TCAGCCATGGCGATTT	RT–qPCR
IRF7	F: CAGTGCTTCTCCAGCACAAA R: TGCATGTGGTATTGCTCGAT	RT–qPCR
STING	F: TGACCGAGAGCTCCAAGAAG R: CGTGGCAGAACTACTTTCAG	RT–qPCR
TBK1	F: AAGAAGGCACACATCCGAGA R: GGTAGCGTGCAAATACAGC	RT–qPCR
NF-κB	F: CATTGCCAGCATGGCTACTAT R: TTCCAGTTCCCGTTTCTTCAC	RT–qPCR
MDA5	F: CAGCCAGTTGCCCTCGCCTCA R: AACAGCTCCCTTGCACCGTCT	RT–qPCR
LGP2	F: CCAGAATGAGCAGCAGGAC R: AATGTTGCACTCAGGGATGT	RT–qPCR
IFN-α	F: ATGCCACCTTCTCTCACGAC R: AGGCGCTGTAATCGTTGTCT	RT–qPCR
IFN-β	F: ACCAGGATGCCAACTTCT R: TCACTGGGTGTTGAGACG	RT–qPCR
IFN-γ	F: ATCATACTGAGCCAGATTGTTTCG R: TCTTTCACCTTCTTCACGCCAT	RT–qPCR
GAPDH	F: GCACTGTCAAGGCTGAGAACG R: GATGATAACACGCTTAGCACCAC	RT–qPCR

**Table 2 viruses-16-00330-t002:** GenBank accession numbers of the IFITM3 genes used in this study.

Name of Species	GenBank Accession Number
*Homo sapiens*	BC070243.1
*Gorilla gorilla gorilla*	KU570011.1
*Capra hircus*	KM236557.1
*Gallus gallus*	KC876032.1
*Serinus canaria*	XM_009102512.1
*Anas platyrhynchos*	KJ739866.1
*Mus musculus*	BC010291.1
*Anser cygnoides*	KX594327.1
*Sus scrofa*	JQ315416.1

**Table 3 viruses-16-00330-t003:** siRNA sequences targeting the IFITM3 gene.

siRNA	Sequences	Sequences
siIFITM3-35	GCAUCAACAUGCCUUCUUATT	UAAGAAGGCAUGUUGAUGCTT
siIFITM3-200	GGAUCAUCGCCAAGGACUUTT	AAGUCCUUGGCGAUGAUCCTT
siIFITM3-242	GGACAGCGAAGAUCUUUAATT	UUAAAGAUCUUCGCUGUCCTT
siNC	UUCUCCGAACGUGUCACGUTT	ACGUGACACGUUCGGAGAATT

## Data Availability

The original contributions presented in the study are included in the article, and further inquiries can be directed to the corresponding author.

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
