# Peer review of "Analysis of Chicken IFITM3 Gene Expression and Its Effect on Avian Reovirus Replication"

_viruses, 2024, doi:10.3390/v16030330_

Round 1
Reviewer 1 Report
Comments and Suggestions for Authors
The authors investigate the avian homolog of the IFITM3 gene, both through bioinformatic/phylogenetic analysis and by examining tissue expression by quantitative PCR. The authors use an in vitro system in immortalized chicken embryonic fibroblasts to visualize cell localization of tagged chicken IFITM3, and use an overexpression/knockdown approach to examine the effect of IFITM3 on avian reovirus replication. The authors identify an antiviral function to IFITM3, similar prior research involving other avian viruses and/or animal hosts. Finally, the authors examine the effect of IFITM3 overexpression/knockdown on the expression of innate immune genes following avian reovirus expression, and observe modest differences in some of the genes.
Overall, the primary issue with the manuscript is one of novelty/significance. Much of the information on avian IFITM3 has already been reported by previous studies, in one form or another, or is confirmatory of effects previously observed in other model systems. There are some technical issues with how the quantitative RT-PCR was performed that should be addressed, as well as some formatting issues with some of the figures that could aid in their clarity, as described below:
Fig. 5 (and Fig. 7): The figure indicates that 3 replicates were performed to generate the data, but more information is needed to interpret the results. Are these biologic replicates (i.e. 3 different animals were sacrificed and each of the organs was separately processed for RT-PCR), or are they technical replicates (i.e. one biological sample was measured three times to assess within-assay variability). Typically, one would use biologic replicates to assay different samples, but have multiple technical replicates as well to account for within-assay differences.
Fig. 6: It would be nice to have some means of assessing, at the protein level, how much the overexpression system led to differences in IFITM3 above the endogenous amount – I understand that there is not likely a chicken-specific IFITM3 antibody available commercially, but given the homology in the CD225 domain, there might be enough cross-reactivity with another IFITM3 antibody to make this assessment.
Fig. 6D and G: the Y-axis is misleading...the data should be plotted on a log scale to better assess the scope of the inhibition/enhancement of replication. A column graph may also be a better choice than a line graph. Additionally, it would be helpful to show more than one timepoint, so the kinetics of replication could be better assessed.
Fig. 6B/E (and elsewhere): it is not clear how the RT-PCR data is normalized. What does a value of “100” mean – is this the percentage of the amount observed in the negative control? An arbitrary value? Typically, using a delta-deltaCT method, the data would be normalized first to a housekeeping gene (i.e. GAPDH, as was done) but then normalized a second time to a reference value (i.e. levels in untransfected, uninfected cells – the “second delta”), which would result in that control having a value of “1” (or “100”). For example, it would appear in figure 5 that the results were normalized to those found in the heart, so that you could interpret that levels in the thymus are ~5-fold higher. This is not the case for the RT-PCR shown in Fig. 6C or E, or in Figure 7. More explanation is needed.
Line 206-7 – a citation is necessary for the “previous homology analysis”
Line 249 – please clarify “Table 3. on ARV infection”
Line 291 – the authors claim that “The expression of MAVS, IRF7, STING, NF-κB, MAD5, LGP2, IFN-α and IFN-β was upregulated after infection regardless of whether IFITM3 was overexpressed or inhibited” but this statement is not justified by the data provided, since no data was shown for uninfected cells (the only comparison was between IFITIM3 overexpression or not). Either show data for uninfected cells, indicate that the levels were normalized to uninfected cells (i.e. uninfected cells would be a level of “1” and infected cells had ~100-fold more mRNA (unlikely)), or revise this sentence to better reflect the data presented.
Comments on the Quality of English LanguageThere are some minor grammatical errors that could be corrected with some copy editing, but overall the English is acceptable.
Author Response
Response: We greatly appreciate your professional review of our article. As you noted, the previous version of the manuscript had several problems that needed to be addressed. We have made corrections to our previous draft according to your suggestion, and the detailed corrections are listed below.
1.Fig. 5 (and Fig. 7): The figure indicates that 3 replicates were performed to generate the data, but more information is needed to interpret the results. Are these biologic replicates (i.e. 3 different animals were sacrificed and each of the organs was separately processed for RT-PCR), or are they technical replicates (i.e. one biological sample was measured three times to assess within-assay variability). Typically, one would use biologic replicates to assay different samples, but have multiple technical replicates as well to account for within-assay differences.
Response: Thank you for this valuable question. The data in this study were obtained from biological replicates and technical replicates. The data were derived from three independent animal experiments. In addition, each sample was measured three times during real-time fluorescence quantitative PCR (RT‒qPCR). This part of the manuscript has been described in detail (lines 184-185).
2.Fig. 6: It would be nice to have some means of assessing, at the protein level, how much the overexpression system led to differences in IFITM3 above the endogenous amount – I understand that there is not likely a chicken-specific IFITM3 antibody available commercially, but given the homology in the CD225 domain, there might be enough cross-reactivity with another IFITM3 antibody to make this assessment.
Response: Thank you for this valuable comment. Your suggestion is good, and it is necessary to assess the extent to which the overexpression system led to differences in IFITM3 above the endogenous amount at the protein level. We also considered purchasing commercial avian IFITM3 monoclonal antibodies to study the protein expression of IFITM3. At present, most commercially available IFITM3 antibodies are mammalian antibodies of human, rabbit, or mouse origin, and there are no commercial avian IFITM3 antibodies available for sale.
The CD225 domain is a highly conserved region of amino acids shared by members of the CD225 superfamily[1]. Previous studies have shown that the CD225 domain contains multiple key regions associated with antiviral effects, which are also closely correlated with the antiviral effects of IFITM3[2]. It is speculated that the homology of the CD225 domain may cause IFITM3 proteins derived from different species to exhibit certain similarities in their biological functions. Your advice is very good. In subsequent studies, we will purchase commercial antibodies from other species to test for cross-reactivity. In addition, monoclonal antibodies against chicken IFITM3 will be prepared to study IFITM3 overexpression at the protein level. We will also conduct more in-depth researches on the antiviral mechanism of IFITM3 in ARV infection.
1.John, S.P.; Chin, C.R.; Perreira, J.M.; Feeley, E.M.; Aker, A.M.; Savidis, G.; Smith, S.E.; Elia, A.E.; Everitt, A.R.; Vora, M. et al. The CD225 domain of IFITM3 is required for both IFITM protein association and inhibition of influenza A virus and dengue virus replication. J. Virol. 2013, 87, 7837-7852, doi:10.1128/JVI.00481-13.
2.Kim, Y.C.; Jeong, M.J.; Jeong, B.H. Genetic characteristics and polymorphisms in the chicken interferon-induced transmembrane protein (IFITM3) gene. Vet. Res. Commun. 2019, 43, 203-214, doi:10.1007/s11259-019-09762-y.
3.Fig. 6D and G: the Y-axis is misleading... the data should be plotted on a log scale to better assess the scope of the inhibition/enhancement of replication. A column graph may also be a better choice than a line graph. Additionally, it would be helpful to show more than one timepoint, so the kinetics of replication could be better assessed.
Response: Thank you for the suggestion. We have revised Figs. 6D and G in the manuscript.
4.Fig. 6B/E (and elsewhere): it is not clear how the RT-PCR data is normalized. What does a value of “100” mean – is this the percentage of the amount observed in the negative control? An arbitrary value? Typically, using a delta-deltaCT method, the data would be normalized first to a housekeeping gene (i.e. GAPDH, as was done) but then normalized a second time to a reference value (i.e. levels in untransfected, uninfected cells – the “second delta”), which would result in that control having a value of “1” (or “100”). For example, it would appear in figure 5 that the results were normalized to those found in the heart, so that you could interpret that levels in the thymus are ~5-fold higher. This is not the case for the RT-PCR shown in Fig. 6C or E, or in Figure 7. More explanation is needed.
Response: Thank you for this valuable question. In this study, we used real-time PCR (Thermo Fisher Scientific, QuantStudio 5) for detection with GAPDH as the housekeeping gene. The data were analyzed by the 2-ΔΔCt method and graphed using GraphPad Prism 8. The analysis of the data was referenced from the research of Li and Wu [1, 2]. First, we normalized the data to GAPDH, and then the data were normalized a second time to the control group. Fig 6B shows the RT‒qPCR results for DF-1 cells with high IFITM3 expression. The data were normalized to those of the control group transfected with pEF1α-Myc (Vec), for which the value was 1. The results showed that, compared with that in the control group, the expression of IFITM3 in D-F1 cells was significantly upregulated, and its expression was increased by approximately 155-fold. Figure 6C shows the effect of IFITM3 overexpression on ARV replication in DF-1 cells. Similarly, the data were normalized to those of the control group transfected with pEF1α-Myc (Vec) at a value of approximately 100. The results showed that the viral load of ARV was significantly reduced after IFITM3 overexpression. Fig. 6E and 6F show the effect of inhibiting IFITM3 expression on ARV replication in DF-1 cells, and the data were normalized to the control transfected with siNC at a value of approximately 100. The results showed that the level of ARV replication was significantly increased after the IFITM3 expression was inhibited.
1.Li X, Jia Y, Liu H, et al. High level expression of ISG12(1) promotes cell apoptosis via mitochondrial-dependent pathway and so as to hinder Newcastle disease virus replication[J]. Vet Microbiol, 2019,228:147-156.DOI:10.1016/j.vetmic.2018.11.017.
2.Wu X, Liu K, Jia R, et al. Duck IFIT5 differentially regulates Tembusu virus replication and inhibits virus-triggered innate immune response[J]. Cytokine, 2020,133:155161.DOI:10.1016/j.cyto.2020.155161.
5.Line 206-7 – a citation is necessary for the “previous homology analysis”
Response: Thank you for your comment. In this study, DNAStar and MEGA software were used for homology analysis and phylogenetic tree construction. Homology analysis revealed that chicken IFITM3 exhibited the highest homology (99.4%) with that of Anas platyrhynchos. The homologies between chicken IFITM3 and those of Anser cygnoides and Serinus canaria were 46% and 45.7%, respectively, and the homologies between chicken IFITM3 and those of Homo sapiens, Gorilla gorilla gorilla, Capra hircus, Sus scrofa, and Mus musculus were 50.4%, 53.5%, 51%, 49%, and 46.5%, respectively. The results of the phylogenetic tree showed that chicken IFITM3 is most closely related to IFITM3 in A. platyrhynchos. A. cygnoides and S. canaria were found in the same group of birds as chickens, but their IFITM3s were distantly related to chicken IFITM3. Chicken IFITM3 is most distantly related to IFITM3s in mammals, such as H. sapiens and G. gorilla gorilla. The results of homology analysis were consistent with the results of the phylogenetic tree. We have revised this term in the manuscript, and the sentence has been changed to "these results are consistent with the results of the homology analysis described above"(lines 210-211).
6.Line 249 – please clarify “Table 3. on ARV infection”
Response: Thank you for your suggestion. Due to reformatting, the manuscript was garbled. We have revised this part of the manuscript (line 253).
7.Line 291 – the authors claim that “The expression of MAVS, IRF7, STING, NF-κB, MAD5, LGP2, IFN-α and IFN-β was upregulated after infection regardless of whether IFITM3 was overexpressed or inhibited” but this statement is not justified by the data provided, since no data was shown for uninfected cells (the only comparison was between IFITIM3 overexpression or not). Either show data for uninfected cells, indicate that the levels were normalized to uninfected cells (i.e. uninfected cells would be a level of “1” and infected cells had ~100-fold more mRNA (unlikely)), or revise this sentence to better reflect the data presented
Response: We thank the reviewer for this comment. In this study, the recombinant plasmid pEF1α-Myc-IFITM3 was transfected into DF-1 cells to overexpress the IFITM3 protein, and cells transfected with the pEF1α-Myc empty vector were used as controls to investigate the role of IFITM3 overexpression in innate immunity against ARV infection. In addition, we designed small interfering RNAs (siRNAs) targeting the chicken IFITM3 gene to inhibit its expression. The siRNAs were transfected into DF-1 cells, and siNC-transfected cells were treated as controls to study the effect of inhibiting IFITM3 expression on innate immunity against ARV infection. We have revised this term in the manuscript. This sentence has been changed to state that after infection, the expression of MAVS, IRF7, STING, NF-κB, MAD5, LGP2, IFN-α and IFN-β was upregulated compared with that in the control group, regardless of whether IFITM3 was overexpressed or inhibited (lines 297-300;lines 377-380).

Reviewer 2 Report
Comments and Suggestions for Authors
Overall the manuscript describes a study on IFITM3 in chicken cell lines and in an animal model. The manuscript reported IFITM3 may exert a negative regulatory effect on TBK1 and IIFN-γ. The effect on IFITM3 has yet to be tested on ARV. While the role of IFITM3 has been determined in influenza and lyssaviruses, there are no publications relating ARV with IFTMS3 yet. Furthermore, it demonstrates that IFITM3 also plays a role in dsRNA viruses infection. The methodology is fine and the conclusion is consistent with the evidence. I felt that the manuscript required minor revision before being accepted. Below are the comments:
1) Figure 4. Subcellular localization of the IFITM3 protein was observed in 63x magnification. This is not an overall representation of the over-expression of IFITM3 in DF1 cells. Besides, qPCR to quantify the relative over-expression, microscopy magnification at 10X or 20X should be provided to represent the spatial over-expression of IFITM3
Author Response
Response: Thank you very much for your professional review of our article. We have carefully considered all your suggestions and made revisions to address your concerns. In the remainder of this letter, we discuss each of your comments and provide corresponding responses.
- Figure 4. Subcellular localization of the IFITM3 protein was observed in 63x magnification. This is not an overall representation of the over-expression of IFITM3 in DF1 cells. Besides, qPCR to quantify the relative over-expression, microscopy magnification at 10X or 20X should be provided to represent the spatial over-expression of IFITM3
Response: Thank you for this valuable comment. The antiviral function of proteins is closely related to their subcellular localization in cells. To determine the localization of IFITM3 in cells, we performed a subcellular localization assay. The cells transfected with the pEF1α-Myc-IFITM3 for 24 hours and 48 hours were collected respectively for testing. Nuclei were stained with DAPI (blue), and the IFITM3 protein was labeled with Alexa Fluor 488-labeled goat anti-mouse IgG (green); imaging was performed using a laser confocal scanning microscope (LEICA-TCS-SP8MP). To obtain higher resolution images, we used a high-numerical-aperture oil lens (usually 63x) for observation. The results showed that the IFITM3 protein was localized in the cytoplasm, and there was no significant difference between the cells transfected for 24 hours and the cells transfected for 48 hours; therefore, we only showed the results of transfection for 24 hours in the manuscript.
Your suggestion is good. In previous studies, the overexpression of IFITM3 was verified by an immunofluorescence assay (IFA), but these data were not presented in the manuscript due to the structure and length of the article.

Round 2
Reviewer 1 Report
Comments and Suggestions for Authors
In the authors response, they indicate that for qPCR, "the data were derived from three independent animal experiments. In addition, each sample was measured three times during real-time fluorescence quantitative PCR." However, in the revised manuscript they only state "The data were obtained from biological replicates and technical replicates." (Lines 184-185). The more specific details, including the number of technical replicates, provided in the author's response should be included.
Author Response
Response: Thank you very much for your professional review of our article. We have carefully considered your suggestions and made revisions to previous manuscript to address your concern. In the remainder of this letter, we provide a corresponding response to your comment.
- In the authors response, they indicate that for qPCR, "the data were derived from three independent animal experiments. In addition, each sample was measured three times during real-time fluorescence quantitative PCR." However, in the revised manuscript they only state"The data were obtained from biological replicates and technical replicates." (Lines 184-185). The more specific details, including the number of technical replicates, provided in the author's response should be included.
Response: Thank you for this valuable comment. We have revised this part in the manuscript, and the sentence has been changed to "The data were obtained from biological replicates and technical replicates. The results are expressed as the mean ± standard deviation (SD) of three independent experiments. And each sample was measured three times during RT‒qPCR"(lines 183-185).
